# Continuous Fiber-Reinforced Aramid/PETG 3D-Printed Composites with High Fiber Loading through Fused Filament Fabrication

**DOI:** 10.3390/polym14020298

**Published:** 2022-01-12

**Authors:** Sander Rijckaert, Lode Daelemans, Ludwig Cardon, Matthieu Boone, Wim Van Paepegem, Karen De Clerck

**Affiliations:** 1Department of Materials, Textiles and Chemical Engineering (MaTCh), Ghent University, Technologiepark-Zwijnaarde 46/70A/130, B-9052 Zwijnaarde, Belgium; sander.rijckaert@ugent.be (S.R.); lode.daelemans@ugent.be (L.D.); Ludwig.Cardon@UGent.be (L.C.); Wim.VanPaepegem@UGent.be (W.V.P.); 2UGCT-Department of Physics and Astronomy, Ghent University, Proeftuinstraat 86/N12, B-9000 Gent, Belgium; Matthieu.Boone@ugent.be

**Keywords:** additive manufacturing, FFF, FDM, mechanical testing, microscopy, low-cost

## Abstract

Recent development in the field of additive manufacturing, also known as three-dimensional (3D) printing, has allowed for the incorporation of continuous fiber reinforcement into 3D-printed polymer parts. These fiber reinforcements allow for the improvement of the mechanical properties, but compared to traditionally produced composite materials, the fiber volume fraction often remains low. This study aims to evaluate the in-nozzle impregnation of continuous aramid fiber reinforcement with glycol-modified polyethylene terephthalate (PETG) using a modified, low-cost, tabletop 3D printer. We analyze how dimensional printing parameters such as layer height and line width affect the fiber volume fraction and fiber dispersion in printed composites. By varying these parameters, unidirectional specimens are printed that have an inner structure going from an array-like to a continuous layered-like structure with fiber loading between 20 and 45 vol%. The inner structure was analyzed by optical microscopy and Computed Tomography (µCT), achieving new insights into the structural composition of printed composites. The printed composites show good fiber alignment and the tensile modulus in the fiber direction increased from 2.2 GPa (non-reinforced) to 33 GPa (45 vol%), while the flexural modulus in the fiber direction increased from 1.6 GPa (non-reinforced) to 27 GPa (45 vol%). The continuous 3D reinforced specimens have quality and properties in the range of traditional composite materials produced by hand lay-up techniques, far exceeding the performance of typical bulk 3D-printed polymers. Hence, this technique has potential for the low-cost additive manufacturing of small, intricate parts with substantial mechanical performance, or parts of which only a small number is needed.

## 1. Introduction

In recent years, additive manufacturing (also known as 3D printing) has started to gain a foothold in production processes for the aerospace [1,2], automotive [3,4] and medical fields [5,6], among others [7], since they allow for the construction of complex structures without the need for additional molding or assembly. Extrusion-based additive manufacturing (Fused Filament Fabrication, FFF) has been widely successful in providing a quick, straightforward way of producing models, prototypes, and non-loadbearing parts, but they lack the mechanical properties to be used in structural applications [8,9,10]. The introduction of short fiber reinforcement can improve the modulus of these parts by over 400%, but the inclusion of discontinuous fibers provides initiation sites for fracture, causing the strength and impact resistance to fall behind [11,12,13].

New developments in the field of FFF printing have allowed for the production of continuous fiber-reinforced 3D-printed parts [14,15,16]. Several methods have been proposed for the inclusion of continuous fiber reinforcement into the printing process. These methods include impregnation of the fibers before entering the printing head, a process also used in currently available commercial systems such as the Markforged MarkTwo [17,18,19,20], impregnation of the fibers after the polymer leaves the printing head [21], and impregnation of the fibers inside the printing head [14,22,23]. Although the printing technique using pre-impregnated fibers shows similarities to better known Automated Tape Laying (ATL) methods for traditional composites, the latter technology is meant for larger parts and requires far more expensive equipment [24].

Three-dimensionally printed parts are regarded as anisotropic, and this may even be more so for continuous fiber-reinforced 3D-printed parts. Thus, an evaluation under various loading modes is crucial to establish the knowledge needed to characterize these materials. Several authors have already characterized the stiffness and strength properties of 3D-printed composites, but the overall fiber loading of the samples produced in these studies remained low (typically below 15 vol%), and mechanical characterization is usually limited to a single loading mode in the fiber direction [14,15,18,25,26]. To gain a more complete picture of the capability of this novel material type, the spectrum of load cases and fracture mechanics needs to be expanded. Additionally, microscopic analysis of the microstructure of these composites is still lacking and thus knowledge on the effect of printing parameters and fiber content on the inner structure of these materials is missing. However, it is well known that layer height, inter-bead spacing and other dimensional properties have a significant impact on the mechanical performance of non-reinforced 3D-printed structures [8,27,28], and that the ply thickness can largely influence the properties of traditional composites [29,30,31,32]. Therefore, a study on 3D-printed composites needs a more profound microscopic analysis as well, using for example both optical microscopy and computed tomography (µCT). These studies are seeing widespread use for traditional composite materials and have been shown to yield important insights [33,34]. 

Several polymer matrix materials have been reported in literature, ranging from typical 3D printing materials such as PLA (polylactic acid) and ABS (acrylonitrile butadiene styrene) to high engineering polymers such as PEEK (polyether ether ketone). Other polymers often used are PA (polyamide) and PETG (glycol-modified polyethylene terephthalate) [18,26,35,36,37]. PETG is an amorphous polymer widely used in 3D printing which performs well mechanically and is far less brittle compared to PLA and ABS (124% elongation at fracture vs. 6% and 100%, respectively) [38]. Compared to the semi-crystalline polyamide printing polymers, the amorphous PETG shows a lower thermal shrinkage after printing, leading to reduced warping of the printed parts [38]. 

Typical fiber materials used for printed composites are similar to those used in traditional composite production, i.e., carbon fiber, glass fiber, and aramid fiber, as well a natural fibers such as flax [39,40,41]. While aramid fibers are no longer often chosen for traditional composites, their higher abrasion resistance and higher bending strength reduce the risk of fiber fracture during printing while making short turns and tight corners [42,43]. They are therefore more suitable for 3D-printed composites than for traditional methods, where these tight corners are less prevalent.

Published work on continuous aramid fiber-reinforced prints reports tensile moduli up to 9.34 GPa for materials made on a commercially available Markforged printer [18,26] and 3.6 GPa for an in-house built set-up [25]. Data on flexural performance are very limited, with currently only data available on specimens printed with a Markforged commercial printer reaching up to 4.6 GPa [26]. These results thus show that an improved mechanical performance is achieved for 3D composites compared to their non-reinforced counterparts, with reported increases of tensile strength and modulus of up to 270% (164 MPa vs. 61 MPa) and 625% (6.65 GPa vs. 1.06 GPa) [26]. However, the performance remains low in comparison with traditional aramid fiber-reinforced composites, and requires an improvement of an order of magnitude before they can be considered for structural applications. The performance likely remains limited due to two main factors. At first, there is no dedicated compaction step in the 3D printing process in contrast to most traditional composite production processes [17]. Secondly, the fiber loadings in published literature on aramid fiber-reinforced 3D-printed composites remain low compared to traditional fiber-reinforced composites, typically around 10 vol% [18,19,26]. 

To achieve the full potential of the novel technique of 3D printing composites, a higher degree of freedom for production parameters is needed than what is typically available from commercial machines that have limited print settings available. For that reason, researchers have proposed modifications to a standard open-source 3D printer [14,23,44]. This open-source design allows for the comparison of structures and printing parameters (e.g., line spacing and layer height) which are not typically available when working with commercially available systems. 

In this paper, an in-house modified Monoprice Mini tabletop 3D printer was used to produce continuous fiber-reinforced aramid/PETG 3D-printed composites. These composites were mechanically tested to gain insight into the relation between printing parameters and print quality, internal structure, and mechanical performance. In addition, microscopic analysis of printed composites was performed to determine the impregnation quality, fiber loading, and composite microstructure of specimens. Scanning of samples with µCT was used to gain insights into the distribution of voids inside printed parts. 

## 2. Materials and Methods

### 2.1. Materials

Glycol-modified polyethylene terephthalate filament (ColorFabb NGen Clear, filament made of Eastman Amphora AM3300, Belfeld, The Netherlands, 1.75 mm) and aramid fiber (Unidirectional tow Kevlar^®^49, 46 tex, produced by DuPont, Wilmington, DE, USA) were used as received. 

### 2.2. Methods

#### 2.2.1. Adaptation of 3D Printer to Allow Printing of Continuous Fiber-Reinforced Polymer

Specimens are produced on a (modified) consumer-grade desktop Fused Filament Fabrication (FFF) printer (Monoprice Mini, Monoprice USA, Brea, CA, USA). The printer itself consisted of an XZ-movable extruder system (Bowden extruder for filament feed), a Y-gantry with a heated bed, and a nominal printing volume of 120 × 120 × 120 mm^3^, and was controlled by G-code. The standard hot end has been replaced by an in-house developed in-nozzle pultrusion-based printing head. This head consisted of separate feeds for polymer filament (1.75 mm diameter) and dry fiber tow, connected by a flow channel (Figure 1a). After the dry fiber tow and liquefied polymer come together in the bulk of the head, the polymer-impregnated strand leaves the printing head through a standard disposable nozzle (M6 thread, E3D v6, brass, 0.6 mm diameter). The fiber tow is inserted in a straight line to the printing nozzle to minimize friction and abrasion of the fiber tow at the printing head. Currently, there is no automated thread-cutting mechanism and thus the printing path has to be continuous. Note that the proposed printing head is independent of the type of 3D printer and can be connected to any FFF-based equipment.

#### 2.2.2. Printing of Continuous Fiber-Reinforced Polymer Specimens

Samples were printed at a continuous speed of 150 mm/min, with extruder and bed temperatures of 220 and 60 °C, respectively. These are standard temperatures for non-reinforced 3D printing. However, the speed of printing continuous fiber-reinforced composites is significantly lower than standard FFF 3D printing, typically being around 150 mm/s. The fiber loading was controlled by changing the layer height and width, thus controlling the size of one laid-down strand. The polymer filament feed was controlled using GCode commands and ensured that the liquefying chamber was constantly filled with polymer during fiber pultrusion. Unidirectional beam specimens were printed similar to traditional non-reinforced FFF printing, i.e., layer by layer. Samples were printed with 10 layers each consisting of 10 lines. After printing, the nozzle is moved upward and the pultruded strand is cut to remove the specimen from the printing bed.

#### 2.2.3. Determination of Fiber Loading

Fiber loading and void content were determined by measuring the density of samples according to ISO/DIS 1183–1 method A (immersion in distilled water), using a laboratory-scale Precisa XR 205SM-DR and determining the mass fraction of the fibers and matrix by thermogravimetric analysis (TGA) (STA 449 F3 Jupiter, Netzsch, Selb, Germany). The samples were heated from 40 to 800 °C at a heating rate of 20 °C/min, under an oxygen atmosphere (50 mL/min) in a Pt-Rh pan.

#### 2.2.4. Mechanical and Microscopic Analysis

Tensile testing of the printed composites in the fiber direction (0° tensile test) was performed according to ASTM D3039 on an Instron 3369, using wedge clamps and a 50 kN load cell. The specimens had nominal dimensions of 100 × 4.5 ± 0.5 × 3.5 ± 0.5 mm^3^ (L × W × T, where the width and thickness depended on fiber loading). The tensile strain was measured with an Instron Static Axial Clip-on Extensometer (model nr. 2630-106, gauge length 25 mm). The extensometer was removed after initial deformation to prevent damage upon fracture of the specimens. Samples intended for loading until fracture were fitted with aluminum tabs to prevent slip and clamp fracture. 

Flexural testing was performed according to ASTM D790 on an Instron 3369 machine, using a three-point bending setup with 3 mm rollers and a 2 kN load cell. A nominal span-to-thickness ratio of 18/1 was used, on samples with the same dimensions as 0° tensile tests. Tests were performed at a constant displacement speed of 1 mm/min.

Tensile testing of samples in the direction perpendicular to the fiber direction (90° samples) was performed on a dedicated tensile stage (Tensile Sample Holder for Phenom XL, 150 N load cell, produced by Deben UK Limited, Woolpit, UK), with nominal sample sizes of 17.5 × 4 × 2 ± 0.2 mm (L × W × T). Tests were performed at a constant displacement speed of 0.5 mm/min.

Mechanical properties of individual fibers were measured according to ISO 5079 on a dedicated single fiber testing machine (Textechno Favimat™, force resolution 0.01 mN).

The glass transition temperature of both reinforced and non-reinforced printed PETG parts was determined according to ASTM D3418 on a TA instruments DSC Q2000. Samples with a nominal mass of 10 mg were loaded in TZero aluminum pans. A heat–cool–heat cycle ranging from 30 to 240 °C was performed with a heating speed of 10 °C/min. The results were interpreted using TRIOS software developed by TA instruments.

To verify the composite microstructure and to check the fiber loading, optical microscopy was performed on an Olympus BX-51 optical microscope equipped with an Olympus UC30 camera, and electron microscopy was performed on a Phenom XL Desktop SEM. Fiber loading of the fiber-rich areas was determined via analysis of microscopy images of the XY plane in ImageJ. A threshold greyscale value was visually determined to calculate the relative amount of pixels correlating with fiber area. Assuming the microstructure is constant throughout the length of the sample, this relative amount is equal to the fiber volume loading.

#### 2.2.5. X-ray Microtomography (µCT)

High-resolution X-ray tomography (µCT) was performed at the in-house developed HECTOR scanner of the Ghent University Centre for X-ray Tomography (UGCT) [45]. Using a geometrical magnification factor of 50, a reconstructed voxel size of 4 µm was achieved. At the used source settings (90 kVp, 10 W target power), the influence of the focal spot on the spatial resolution is limited. Covering an angular range of 360°, 2401 projections are acquired at an exposure time of 1000 ms per projection image. The raw data are processed and reconstructed to a 3D volume using Octopus Reconstruction. No phase retrieval was applied to maintain maximal contrast at the small voids. Three-dimensional rendering is performed using VGStudioMax 3.4 (Volume Graphics GmbH, Heidelberg, Germany).

## 3. Results and Discussion

### 3.1. Development of Continuous Fiber-Reinforced Printing Set-Up

A consumer-grade desktop 3D printer was modified to allow for high loading with continuous aramid fibers. These modifications are similar to those proposed by Matsuzaki et al. [14]. To achieve a higher fiber loading, tension in the reinforcement fiber had to be lowered so fiber fracture could be prevented. This was done by ensuring a straight path for the fiber to pass through the liquefied polymer, as seen in Figure 1a,b. This set-up has the added benefit of already surrounding the fiber bundle with polymer before exiting the nozzle, as seen in Figure 2a, and allows for the simple switching of the polymer and fiber material.

Within each printed layer, the strands were deposited next to each other. At the end of each strand, the printing head reversed its direction. This reversal often caused problems as the prints detached from the print bed when using straight 90° turns (Figure 1b, ‘straight turn’). Two reversal strategies were developed to mitigate this problem by extending the print path slightly, allowing the next line to attach to both the printer bed (or the previous layer) and the printed ‘extension’ (Figure 1, ‘pointed turn’ and ‘angled turn’). The ‘angled turn’ yielded far more consistent results than a normal ‘straight’ turn or a ‘pointed’ turn, resulting in far fewer failed prints and fiber detachments. This is likely due to the fiber bundles being ‘hooked’ behind the previous line. Thus, this technique was used for printing all further specimens discussed in this work. After printing, the extensions were cut, resulting in beam-like continuous aramid fiber-reinforced PETG 3D-printed composite specimens. Visual inspection of the printed specimens showed that the print quality was good (straight fiber bundles) and reproducible.

The fiber loading and dispersion were altered by adjusting the layer height and the line spacing while retaining the same amount of fibers present in the print. Due to the passive pultrusion of the matrix polymer, meaning that the amount of polymer is determined by the available space at the end of the nozzle, printing lines closer together results in a lower amount of polymer present and thus higher fiber loading in the printed beams. 

### 3.2. Correlation of Printing Parameters to Sample Microstructure and Fiber Loading

Figure 2 shows the difference in structure between a filament leaving the printer nozzle (before laying down on the printer bed) and a printed line (here two lines printed on top of each other). The lack of impregnation before laying down the fibers shows that additional compaction is needed to assure quality. As most of the impregnation happens outside of the nozzle (compare Figure 2a,b), smaller layer heights result in additional compaction pressure exerted by the nozzle on the fiber bundle. Nevertheless, the fibers tend to migrate to the top of the printed line due to the tension in the tow. This tension pulls the fibers to the side of the nozzle opposing the printing direction. The bead width of the print is mostly determined by the width of the nozzle, which was 0.8 mm for the specimen shown in Figure 2. 

As can be seen in Figure 2, the fibers are forced to make an angle of 90° when leaving the heated nozzle, while being pulled to the side of it. This can cause friction and abrasion against the heated nozzle, possibly damaging the fibers during printing. To assess whether or not the printing head affects the mechanical properties of the aramid fibers, dry fibers were pulled through the heated printing setup without the presence of matrix polymer. The tensile strength and modulus before and after ‘dry’ printing were determined, with a strength of 3.28 ± 0.14 GPa and 3.11 ± 0.27 GPa and a modulus of 107.5 ± 3.2 GPa and 98.4 ± 7.6 GPa, respectively, before and after printing (see Appendix A). A slight decline in mechanical properties is thus observed after printing, mainly in modulus. This does, however, not significantly deteriorate the performance of the printed composites. The glass transition temperature of the matrix polymer after printing remained similar for specimens with and without aramid fibers (80, 77, and 77 °C for 0, 20, and 45 aramid fiber vol%, respectively, as measured by Differential Scanning Calorimetry) (see Appendix A). 

The layer height and line width can be adjusted to achieve different microstructures and fiber loadings. Figure 3 shows the microstructure of printed specimens for four different sets of printing parameters (the printing parameters for these samples are shown in Table 1). From these, the following relations can be determined:Composites printed with a large layer height and large line spacing (config. A) result in an array-like microstructure of individual fiber bundles encompassed by matrix polymer.By decreasing the line spacing (A→B), the lines start overlapping within a layer. As the layer height is still relatively large, newly printed lines do not merge with the neighboring lines, but instead, get printed ‘on top’. Surrounding gaps are filled with the matrix polymer.By decreasing the layer height (A→C), an increased downward force compacts the individual lines into each other, resulting in neighboring lines that are still partially stacked on top of one another but produce a more defined layered structure similar to the structure in traditional composite laminates.Additional compaction can be achieved by further decreasing the layer height and line spacing (C D). This decreases the interlaminar resin-rich layers considerably (from 138 ± 13 to 88 ± 24 µm), resulting in higher fiber volume fractions of the specimens. Additionally, large interlayer voids are eliminated.

Hence, by changing the line width and layer height, it is possible to obtain relatively high fiber loadings and a layered microstructure within the printed composites, which both should translate into a better mechanical performance of the specimens. Indeed, the total volume fraction can be tuned between 20 and 45 vol% for the configurations considered here, which is in line with the volume fractions expected from lower-end traditional composite production processes such as hand lay-up. The void content lowers significantly when printing lines closer together, and this effect seems more pronounced when lowering the layer height. It can be concluded that the added pressure exerted by the nozzle on the fiber bundle when printing at lower layer height forces the liquefied polymer more in between the individual fibers, while the closer printing of lines also closes the larger interstitial voids. Note that the voids are not visible on the images taken via optical microscopy. This is likely caused by the grinding and polishing step during sample preparation, which smears the soft thermoplastic matrix and closes the voids at the surface, thus hiding them from view.

Images at higher magnification (Figure 3, bottom row) show that the microstructure consists of both resin-rich and fiber-dominated areas. The fiber volume fraction within the fiber-dominated areas is rather high compared to the total fiber volume fraction within the specimen. This is indeed confirmed by calculations of the volume fraction of resin-dominated areas and the fiber volume fraction in these fiber-dominated areas (see Table 1). While the total fiber volume fraction differs from 22 to 45 vol% for the configurations considered here, the fiber volume fraction within the fiber-dominated areas changes from 56 to 74 vol%. This shows that the reduction in layer height not only lowers the total amount of polymer present per printed line but also decreases the polymer present between individual fibers. This is likely due to a higher amount of liquefied polymer being pushed out of the fiber tows at lower layer height. It can be concluded that it is not desirable to push for too high fiber contents in these materials, since there will be a drop in impregnation quality at higher fiber loading.

The second row of Figure 3 shows that the shapes of the fiber-rich areas in configurations A and B have a distinct irregular outline, as if two fiber tows are stacked together. This is due to the movement of the fiber tow inside the printer nozzle, causing the fiber bundles to not be perfectly centered in the printed line. The fibers tend to be pulled towards the side of the line laid down right before, the placement of the fiber-rich area depends on whether the line was deposited in the positive or negative y-direction. The edges of fiber bundles can protrude from the side of the printed line, causing subsequent bundles to be partially deposited on top of previous lines. This effect is even more pronounced when lowering the line spacing (A→B), where almost the entire bundle is laid down on top of the previous line. 

### 3.3. Analysis of Void Content via µCT

X-ray computed tomography (µCT) was used to visualize the void content in a 3D-printed aramid/PETG composite with 45 vol% fiber loading, corresponding to configuration D in Section 3.3 (Figure 4). This analysis showed that there is a larger void content than what can be determined from imaging via optical microscopy. It is likely that during sample preparation for optical microscopy, the grinding and polishing of the soft thermoplastic matrix closes some of the voids, so that they are no longer observable.

The µCT images show that voids are mainly situated in fiber-rich areas, and at interfaces between individual printed lines within a plane (same height). An average void content of around 5% was present for the 45 vol% aramid fiber specimen (D). This can be compared to traditional composites, where the void content can range between close to 0% and 10% depending on the production method and pressure applied during resin curing [46]. Typical values for composites produced via a hand lay-up process range from 5% to 10% void content, while specimens produced via Resin Transfer Molding (RTM) can go well below that and achieve void contents as low as 0.8% [47]. Using these values as a comparison shows that the 3D-printed composites presented in this work fall well within the range of traditional composites. It can, however, be noted that, in order to produce materials for use in highly demanding sectors (e.g., aerospace engineering), the impregnation quality has to be improved and void content should be lowered.

### 3.4. Mechanical Analysis of 3D-Printed Composites

#### 3.4.1. Tensile Testing

Figure 5 shows the results of tensile testing on aramid/PETG 3D-printed composites with unidirectional fiber loading in the beam direction varying from 0 to 45 vol%. Stress/displacement curves of these tests (Figure 5a) show a significant increase in modulus and strength with rising fiber loading. It can also be seen that the curves do not exhibit a typical straight line typical for continuous fiber-reinforced composites, but have a bilinear behavior. Since the aramid fibers in these composites are not optimally sized for the used polymer matrix, and voids are concentrated at the fiber/matrix interface as can be seen in Figure 4, fibers can more easily detach due to stress, which no longer allows for the efficient transfer of forces to the fibers and causes the modulus to drop. This will also cause an earlier-than-expected fracture.

The tensile modulus of aramid/PETG 3D-printed composite samples with unidirectional fiber loading in the tensile direction varying from 20 to 45 vol% is shown in Figure 5b together with results from published literature and with the Volume Average Rule of Mixture (VARoM) prediction of the tensile modulus. The experimental results show that the tensile modulus of printed composites can reach up to 33 GPa, far greater than what has been reported thus far for aramid-reinforced printed composites. The values reported in literature were mainly limited in mechanical performance due to a lower fiber loading. The achieved modulus remains close to the predicted value with VARoM, indicating that the fibers are aligned closely to the printing direction. These results are comparable to what can be achieved via traditional composite production methods.

The tensile strength in the fiber direction (0°) is shown in Figure 5c, compared to published strengths for traditional aramid/epoxy composites [40,48,49]. While the strength is lower than what can be achieved with traditional composites, the published results show that this can largely be caused by the presence of voids surrounding the fibers and a lack of optimized fiber sizing [49,51,52]. Drops of strength compared to the VARoM ranging between 20% and 60% have been reported in traditional composites [49,51,52]. To the best of our knowledge, an equivalent study has not yet been performed on the necessity of optimized sizing technology for 3D-printed composites.

Figure 5d shows the UD 90° tensile strength of aramid/PETG printed composite samples. The tensile strength of the non-reinforced reference samples is higher compared to the reinforced specimens (42.3 MPa vs. 5 to 11 MPa), indicating that the inclusion of fibers promotes crack formation and failure when tested perpendicular to the fiber direction. This effect can also be seen in traditional composites, with 90° tensile strength, similar to what was reached on this material [50,53]. Failure likely initiates at the fiber/matrix interface since the voids are mainly concentrated around the fiber bundles.

It can be noted that the 90° tensile strength increases when the line spacing is decreased. This can be seen by comparing the values of configurations A to B and C to D, respectively. The closer line spacing introduces more overlap between lines during printing, promoting a better fusion between the printed paths. There is also a significant drop in void content between these configurations. A smaller effect can be noticed comparing configurations A to C and B to D, where a decrease in layer height causes a decrease in 90° tensile strength. This is likely caused by an increase in fiber loading inside the fiber-rich areas, as given in Table 1. The rise in fiber content inside these areas indicates that a smaller amount of matrix polymer is present between the fibers to transfer the forces during testing.

#### 3.4.2. Flexural Testing in Fiber Direction (0°)

Figure 6 shows the results of flexural testing on aramid/PETG 3D-printed composites with unidirectional fiber loading in the beam direction varying from 0 to 45 vol%. Stress/strain curves of these tests (Figure 6a) show a significant increase in modulus and strength with rising fiber loading, while the strain at fracture slightly decreases.

Figure 6b,c show the flexural modulus and strength of these samples, which both increase with fiber loading, similar to the tensile specimens. In comparison to the modulus in tension, however, the modulus under flexural loading is lower, reaching up to 27 GPa. This is likely due to the presence of voids around the fibers and to the detachment of fibers without optimized sizing. The non-uniform distribution of fibers throughout the thickness, with a resin-rich part of layers at the bottom side, likely also contributes to suboptimal mechanical loading of the fibers. Yet, the increase in performance compared to the non-reinforced counterpart is considerable, with high improvements in stiffness (+1650%) and strength (+490%). The performance of the printed specimens is similar to traditional aramid/epoxy composites produced with hand lay-up processes and already exceeds those of 3D-printed aramid-reinforced composites published in literature by a factor of four [26,40]. The 3D printing process, however, also allows for a higher degree of flexibility in the production and design of components.

The flexural strength of aramid/PETG 3D-printed specimens increases linearly with fiber loading, indicating that it has a larger effect on the flexural strength than the specific microstructure of the composite. This is confirmed by comparing the strength of configurations B and C, which have a similar fiber loading but different microstructure. 

From the analysis of the failed specimen in Figure 6d, it can be seen that the predominant mode of failure is layer buckling and fiber detachment at the compressive side, with a distinct lack of fracture at the tensile side of testing. The limited fiber/matrix adhesion due to the presence of voids and lack of sizing results in fiber debonding under shear stresses. Additionally, aramid fibers are prone to buckling at the molecular level under compressive loading, and their compressive strength is far lower than their tensile strength [54]. These reasons likely cause the compressive failure of the specimens.

## 4. Conclusions

In this work, a commercial Monoprice Mini 3D printer was modified with an in-house developed impregnation head that allowed for the production of continuous aramid fiber-reinforced 3D-printed composites. The design allowed for high fiber loadings up to 45 vol%, which is substantially higher than reported so far. 

The line spacing and layer height were altered to change the microstructure of the 3D-printed composite, varying from an array-like structure typical for FFF 3D printing, to a well-defined layered structure similar to those found in traditional composite production. It was shown that lowering the layer height and decreasing the line spacing had a significant influence on the void content of the printed composites, lowering the void content from 16 vol% to 5 vol%. Printing the lines closer together and at a lower height also decreased the presence of voids between the individual lines. The configuration with the highest overall fiber loading only showed voids inside the printed lines, in between the individual fibers.

The mechanical performance of these aramid/PETG 3D-printed composites was investigated both in and perpendicular to the fiber direction. The tensile modulus and strength in the fiber direction increases linearly with fiber loading and yields a highly increased modulus (+1550%) compared to non-reinforced 3D-printed PETG reference materials, and a moderately increased strength (+1150%). The strength increase is lower than expected but is likely due to sub-optimal fiber/matrix bonding caused by the presence of voids at the fiber/matrix interface and a lack of optimized fiber sizing. Tensile strength perpendicular to the fiber direction, however, shows a strong decline compared to the reference, attributed to imperfect fiber impregnation and a lack of optimized fiber sizing for the aramid/PETG interface. It was shown that decreasing the line spacing forces the lines to better fuse together. This caused an increase in tensile strength when tested perpendicular to the fiber direction. Flexural modulus and strength also increase linearly with fiber loading up to +1650% and +490%, respectively. It can be expected that improving the impregnation quality of the reinforcement fibers will further improve these properties. This can be achieved by better compaction of the printed composites, optimized pre-impregnation of the fiber bundles, or dedicated sizing technology for the used fibers. 

Overall, the performance of these materials is comparable to aramid composites produced via traditional composite production methods such as hand lay-up. The developed printing technique is thus a viable option to produce continuous fiber-reinforced composites with high mechanical properties using low-cost FFF 3D printing technology. This technique can be used to provide new production methods for components with difficult geometries, which would otherwise be impossible to achieve with in-mold methods or require the fabrication of very expensive tooling equipment. Using the fundamental knowledge gained in this work on the impact on mechanical performance in combination with novel print paths (e.g., truss-core or woven structures), 3D-printed composites promise to be an ideal material for the future [55,56].

## Figures and Tables

**Figure 1 polymers-14-00298-f001:**
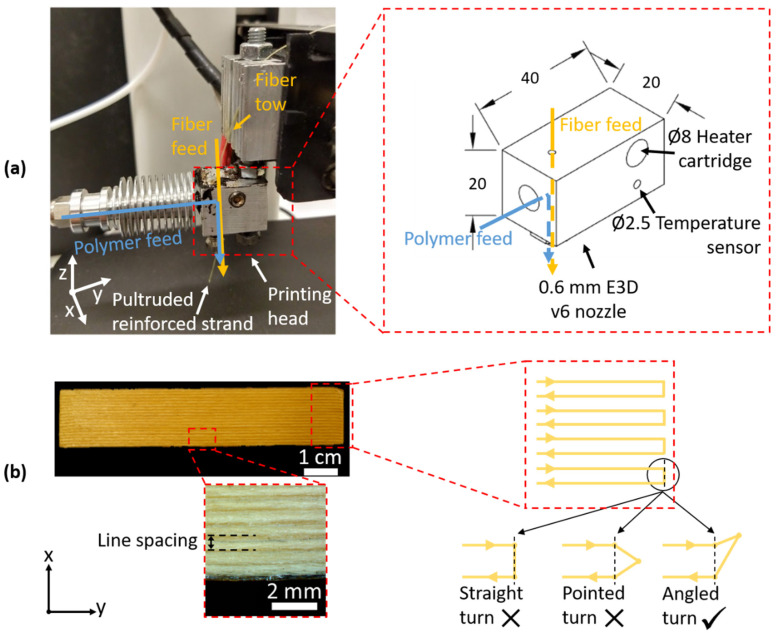
(**a**) This modified tabletop FFF printer with an in-house made in-nozzle impregnation printing head was used to produce 3D-printed composites. In this head, the fibers pass through the melting chamber from a separate feed. (**b**) Continuous aramid fiber-reinforced PETG composite beam specimens were successfully printed by adjusting the cornering path. When cornering at the end of a printed line, fibers often showed detaching, causing the print to fail. An angled turn yielded the most consistent and best results as the fiber bundle could ‘hook’ behind the previous line.

**Figure 2 polymers-14-00298-f002:**
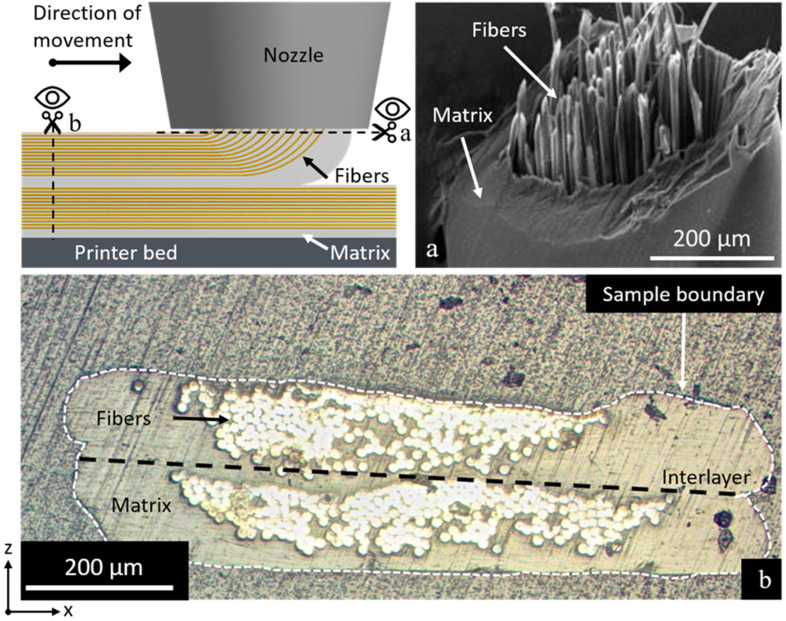
Schematic representation of fiber movement and impregnation during printing. When the fiber tow first exits the nozzle straight down (**a**), the entire tow is surrounded by polymer, but only little impregnation occurs. After being compacted against the printer bed (**b**), the liquefied polymer is forced in between individual fibers. Overflow of polymer is deposited besides the fiber tow. For this image, a printing nozzle with 0.8 mm internal diameter was used, resulting in resin-rich areas at the sides of the printed lines.

**Figure 3 polymers-14-00298-f003:**
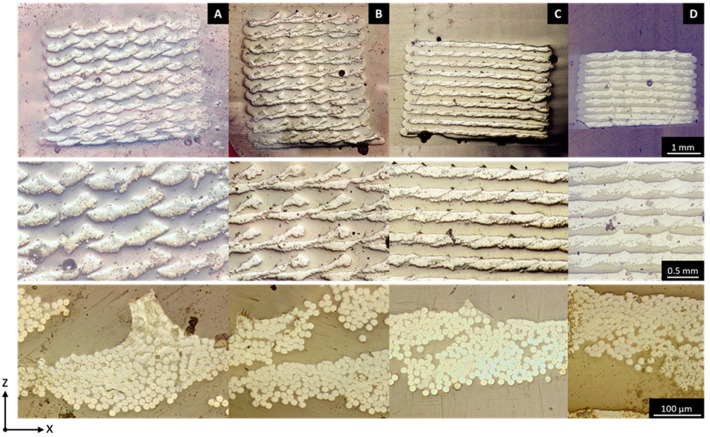
Microscopic analysis of cross-sectional cuts of 20 vol% (**A**), 25 vol% (**B**,**C**) and 45 vol% (**D**) aramid/PETG composites. An evolution towards a continuous fiber-reinforced layer can be seen when going towards higher fiber content where printed lines overlap and the layer height is reduced. Full-size images of the top row can be found in Appendix A.

**Figure 4 polymers-14-00298-f004:**
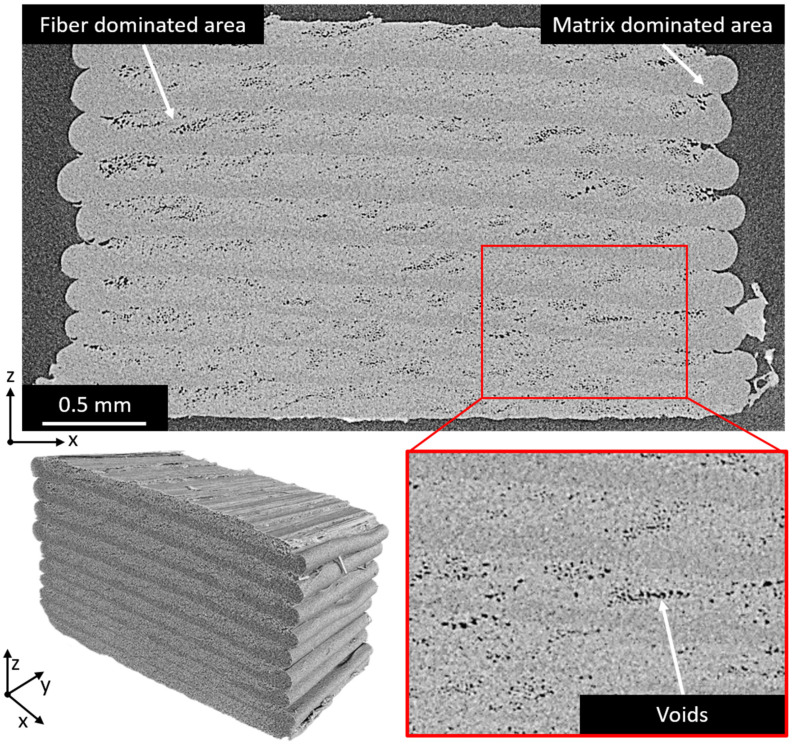
Single slice of a µCT scan and generated 3D image of a 3D-printed aramid/PETG composite with 45% fiber content. These images show the visible void content around fiber-rich areas. A higher concentration of voids can be seen at the sides of the specimen.

**Figure 5 polymers-14-00298-f005:**
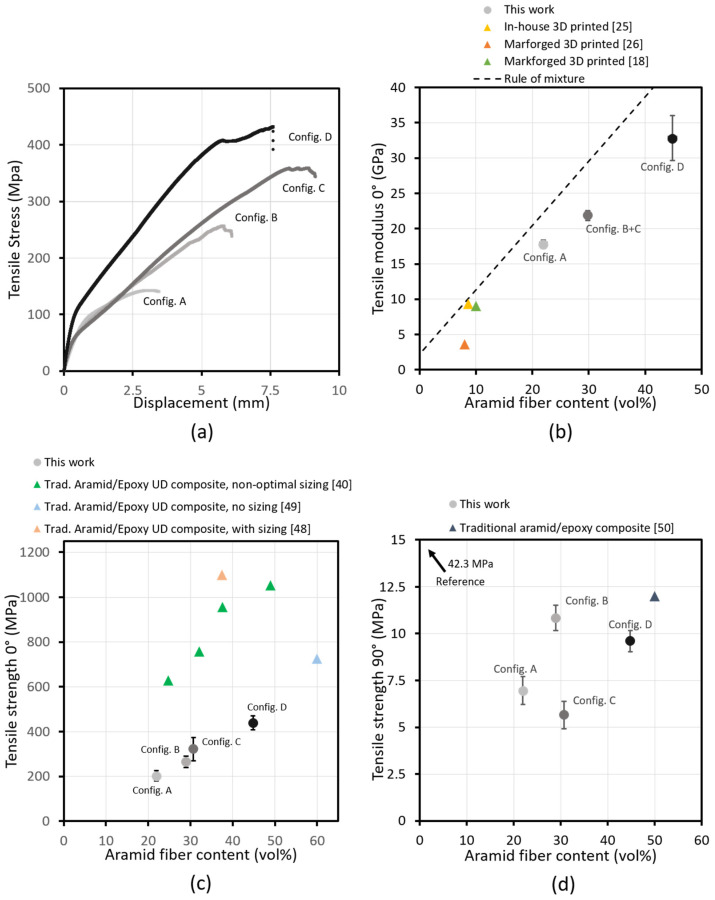
(**a**) Stress/displacement curves of tensile tests on UD 0° aramid/PETG printed composites show that the composites exhibit a bilinear-like behavior, with the modulus decreasing after a certain point. This is likely due to sub-optimal fiber adhesion, caused by high void fraction within the fiber-rich areas and a lack of optimal sizing. The tensile modulus was determined at the initial slope. (**b**) The tensile modulus of UD 0° printed aramid/PETG composites increases linearly with added fiber volume content. Comparison to published works shows high control over fiber content modulation [18,25,26]. (**c**) The tensile strength of UD 0° printed aramid/PETG composites increases linearly with added fiber volume content. Comparison to traditional aramid/epoxy composites shows lower-than-expected strength, and results from literature show that the lack of sizing can cause this drop [40,48,49]. (**d**) The tensile strength of UD 90° printed aramid/PETG composites compared to a non-reinforced 3D-printed PETG sample. A significant drop in tensile strength can be noted when adding continuous fibers perpendicular to the tensile direction. This drop can also be seen with traditional aramid/epoxy composites [50]. Reduction in line width improves the tensile strength.

**Figure 6 polymers-14-00298-f006:**
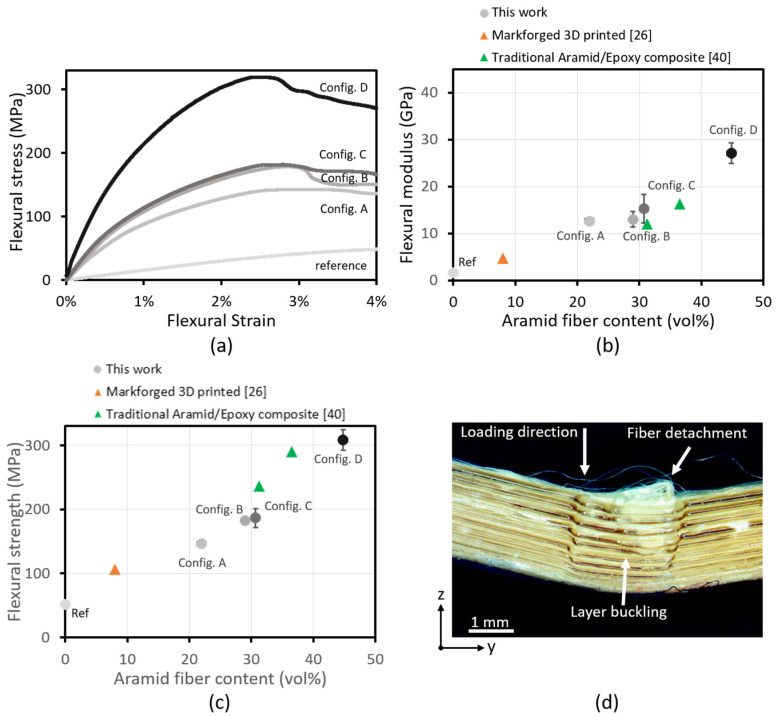
(**a**) Stress/strain curves of 3-point bending tests on UD 0° aramid/PETG printed composites (stopped at 4% strain) show an increase in maximum stress and a slight decrease in strain at max stress for higher fiber content. (**b**) Modulus and (**c**) strength are shown, compared to results from literature [26,40]. A linear increase in modulus and strength with fiber loading can be seen. The performance is comparable to traditional Aramid composites. (**d**) Analysis of a failed specimen shows a large amount of layer buckling and fiber detachment at the compression side.

**Table 1 polymers-14-00298-t001:** Printing parameters for samples shown in Figure 3.

	Layer Height (mm)	Line Spacing (mm)	Fiber Content (vol%)^1^	Void Content (vol%) ^1^	Matrix Dominated Area (vol%) ^2^	Fiber Content in Fiber Dominated Area (vol%) ^2,3^
**A**	0.4	0.45	22	16	60	55.7 ± 1.0
**B**	0.4	0.3	29	11	57.5	66.3 ± 2.5
**C**	0.3	0.4	31	9	55.5	70.9 ± 1.7
**D**	0.25	0.3	45	5	35	73.9 ± 3.6

^1^ Based on density and TGA. ^2^ Based on optical microscopy. ^3^ Calculated using overall fiber loading (TGA) and total area of fiber-rich zones (OM).

## Data Availability

Data is contained within the article or Appendix A.

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
