# Peer review of "Continuous Fiber-Reinforced Aramid/PETG 3D-Printed Composites with High Fiber Loading through Fused Filament Fabrication"

_polymers, 2022, doi:10.3390/polym14020298_

Round 1

Reviewer 1 Report

Dear all,

Greetings and happy new year 2022

please find enclosed our comments regarding this article

- Referenced : polymers-1535167

- Titled: Continuous fiber-reinforced aramid/PETG 3D printed composites with high fiber loading through low-cost Fused Filament Fabrication.

This paper deals with (3D) printing, this technique has huge potential for low-cost additive manufacturing of small important subjects for our daily life. In my opinion this research paper can be accepted for publication in Polymers after minor revisions and after adressing these comments.

- Title: very long

- Abstract: in your case (up to 1650% compared to non-reinforced 3D printed PETG) this phrase need more clarifications

- Keywords: you can add another one

- Introduction:  for the aerospace [X], automotive [Y], medical [Z] and other 
fields [W],  Please put each reference near to the apropriate application for all your introduction!

- Materials and Methods: ok (0° page 4)?

- Results: what are the limitation of your modified tabletop FFF (3D) printer.

- Conclusion: in the abstract something are missing and we found it in the conclusion

- References: please update your references 2021 and 2022 and fellow the journal template for your last reference

With regards

Author Response

Please find our answers in attached file.

Reviewer 2 Report

The peer-reviewed paper presents the results of the thorough study of the proposed 3D printing technology, which uses the in-nozzle impregnation of continuous fiber reinforcement with a modified 3D printer head. The article describes in detail four strategies for laying printed lines, analyzes the microstructure of the resulting composite, their strength and elastic properties under tension and 3-point bending, as well as the volume fraction of fiber. The results of the research are summarized in conclusions (lines 273 - 287), which are very important for practice. The results of continuous fiber-reinforced layer printing at the different layer height and line spacing are clearly illustrated and accompanied by quantitative test results of samples (see Table 1).

The article is well organized and its content is convincingly represented by good scientific style.

The technology presented in the article is very promising, particularly, for the manufacture of composite structures of rather complex geometry. Therefore, it seems useful that the authors, on the basis of the experience gained, give some estimates of the possibility of using the results obtained in 3D printing of real structures.

One error found on line 72

Author Response

(The authors gave the same response as above.)
